# Tryptophan and Cortisol Modulate the Kynurenine and Serotonin Transcriptional Pathway in the Kidney of *Oncorhynchus kisutch*

**DOI:** 10.3390/ani13223562

**Published:** 2023-11-18

**Authors:** Luis Vargas-Chacoff, Daniela Nualart, Carolina Vargas-Lagos, Francisco Dann, José Luis Muñoz, Juan Pablo Pontigo

**Affiliations:** 1Laboratorio de Fisiología de Peces, Instituto de Ciencias Marinas y Limnológicas, Universidad Austral de Chile, Valdivia 5090000, Chile; daniela.nualart@gmail.com (D.N.); franciscojavierdann@gmail.com (F.D.); 2Centro FONDAP de Investigación en Dinámica de Ecosistemas Marinos de Altas Latitudes (IDEAL), Universidad Austral de Chile, Valdivia 5090000, Chile; 3Integrative Biology Group, Valdivia 5090000, Chile; 4Millennium Institute Biodiversity of Antarctic and Subantarctic Ecosystems, BASE, University Austral of Chile, Valdivia 5090000, Chile; 5Escuela de Graduados, Programa de Doctorado en Ciencias de la Acuicultura, Universidad Austral de Chile, Puerto Montt 5480000, Chile; 6Escuela de Medicina Veterinaria, Facultad de Recursos Naturales y Medicina Veterinaria, Universidad Santo Tomás, Puerto Montt 5480000, Chile; carolinavargaslagos@gmail.com; 7Centro i~Mar, Universidad de los Lagos, Puerto Montt 5480000, Chile; joseluis.munoz@ulagos.cl; 8Laboratorio Institucional, Facultad de Ciencias de la Naturaleza, Universidad San Sebastián, Puerto Montt 5480000, Chile; juan.pontigo@uss.cl

**Keywords:** stress, neurotransmitter, cell culture, fish, aquaculture

## Abstract

**Simple Summary:**

Our results indicate activation of the kynurenine pathway and serotonin activity when stimulated with tryptophan and cortisol supplementation. An amount of 95% of tryptophan is degraded by the kynurenine pathway, indicating the relevance of knowing how this pathway is activated and if stress levels associated with fish culture trigger its activation. Additionally, it is essential to know the consequence of increasing KYNA levels in different species in the short and long term, and even during the fish ontogeny.

**Abstract:**

Aquaculture fish are kept for long periods in sea cages or tanks. Consequently, accumulated stress causes the fish to present serious problems with critical economic losses. Fish food has been supplemented to reduce this stress, using many components as amino acids such as tryptophan. This study aims to determine the transcriptional effect of tryptophan and cortisol on primary cell cultures of salmon head and posterior kidney. Our results indicate activation of the kynurenine pathway and serotonin activity when stimulated with tryptophan and cortisol. An amount of 95% of tryptophan is degraded by the kynurenine pathway, indicating the relevance of knowing how this pathway is activated and if stress levels associated with fish culture trigger its activation. Additionally, it is essential to know the consequence of increasing kynurenic acid “KYNA” levels in the short and long term, and even during the fish ontogeny.

## 1. Introduction

Fish in aquaculture suffer high stress levels due to abiotic variables such as temperature, hypoxia, salinity, pH, or eutrophication and biotic variables such as viruses, bacterial, sea lice, or harmful micro-algae blooms [1,2,3]. As a reaction to this event, Wendelaar-Bonga (1997) [3] mentioned that the fish have two axis stress responses, having different actions at different times: the first is for acute stress, “brain-sympathetic-chromaffin” (BSC), and the second axis is for chronic stress, “hypothalamus-pituitary-interrenal” (HPI) [3,4]. Farmed fish are typically kept for a long period in sea cages or tanks. Consequently, accumulated stress causes the fish to present serious problems with critical economic losses. Chronic stress modulates the endocrine response, with cortisol being the final product and used as a stress indicator. This hormone release is controlled by the adrenocorticotropic hormone (ACTH), and its secretion is controlled by corticotropin-releasing factor (CRF) [2]. Cortisol binds with glucocorticoid receptors (GRs) and generates several physiological effects in peripheral tissues in order to overcome stress and recover the pre-stress homeostatic state [3,5,6]. Øverli et al. [7,8] described that neurotransmitters of brain origin, such as noradrenaline (NAd), dopamine (DA), or serotonin (5-HT), are involved in the control and integration of responses to physiological stress in teleost fish [7,9,10,11].

This activity can be traduced into high levels of brain serotoninergic activity after exposure to different stressors. In aquaculture, the fish are exposed to stressors such as handling, pollutants, crowding, diseases, and the presence of ectoparasites (sea lice). These stressors induce high levels of serotoninergic activity, as was observed in several fish species including the coho salmon *Onchorynchus kisutch* [11,12,13]. Several authors indicated that fishes presented high levels of serotonin (5-HT) due to tryptophan supplementation, which reduced cortisol levels and underwent changes, for example, a decrease in aggressive behavior, reduction in cortisol at high temperature, exposure to air, chasing, and high stocking density [14,15,16,17,18,19,20,21,22]. However, all these studies are short-term experiments, and therefore long-term experiments are required [23].

Tryptophan is an essential amino acid for mammals and fish. It is involved in immune tolerance mechanisms mediated by its metabolites, following the enzymatic activity of indoleamine 2,3-dioxygenase in leucocytes [24,25]. Results in European seabass HKL primary cell cultures suggest altered pro-inflammatory signals that counteract the inflammatory response caused by increased tryptophan availability [26]. This amino acid is converted to 5HT through the activity of tryptophan hydroxylase and aromatic L-amino acid decarboxylase in the presence of vitamin B6 [27]. Another pathway is the kynurenine pathway (KP). It is mediated by two rate-limiting enzymes, namely tryptophan 2,3-dioxygenase (TDO) and indoleamine 2,3-dioxygenase (IDO) [28], which metabolizes to kynurenine (KYN) [29], and KYN is metabolized in kynurenic acid by kynurenine aminotransferase 2 (KIAT 2) [30,31,32] (Figure 1). Due to its toxic potential, this pathway may cause stress, which can suppress the immune system, as mentioned by [3]. Kaczorek et al. (2017) [33] indicated that feed supplementation with kynurenic acid in rainbow trout aggravated damage signs in the liver, kidneys, and gills, and the effect was dose-dependent in fish infected with *Yersinia ruckeri*.

In Chile, salmon farming is an important economic activity, with Atlantic salmon (*Salmo salar*) and coho salmon (*Oncorhynchus kisutch*) being the main aquaculture species in Chile, and placed first in coho salmon production worldwide. Generally, coho salmon are less susceptible to sea lice than Atlantic salmon, but they are still affected [11]. The sector is also suffering the effects of climate change due to rising temperatures, low oxygen levels, and harmful micro-algae blooms [2].

Therefore, this study aimed to describe how supplementation with tryptophan and cortisol modulates the neuroendocrine and kynurenic pathway. We used the mRNA transcription of *Tryptophan Hydroxylase* (*TPH2*), *5HT1α receptor*, *INFγ*, and *KIAT 2* as markers in the primary cell culture of the head and posterior kidney of coho salmon (*Oncorhynchus kisutch*) following the pathway (Figure 1).

## 2. Materials and Methods

### 2.1. Animals

We used immature healthy specimens of coho salmon (*Oncorhynchus kisutch*). A total of 12 fish weighing approximately 300 ± 12 g with a length of 39 ± 5 cm, post-smolt stage, were obtained from Metri Station (Universidad de Los Lagos, Osorno, Chile). They were transported to laboratories at the Faculty of Science (Universidad Austral de Chile, Valdivia, Chile) and distributed into seawater (35 psu) tanks (500 L) with a continuous flow-through system, 12:12 h light/dark photoperiod cycle, and a water temperature of 13 ± 2 °C. The fish were acclimated for two weeks to avoid the stress from handling and transportation. During these acclimation and maintenance stages, fish were fed ad libitum using EWOS Transfer 100 pellet feed, without boost. Three fish were sampled, and each fish head kidney and posterior kidney were removed; each tissue had three replicates. All fish were captured, anesthetized with a lethal dose of 2-phenoxyethanol (1 mL/L, Fluka-77699-500ML SIGMA-ALDRICH, SLS Ireland, Dublin, Ireland), and euthanized through spinal sectioning before tissue removal [1,2,3,4,5,6,7,8,9,10,11,12,13,14,15,16,17,18,19,20,21,22,23,24,25,26,27,28,29,30,31,32,33,34].

All experimental protocols complied with guidelines for using laboratory animals, as established by the Chilean National Commission for Scientific and Technological Research (ANID), the Universidad Austral de Chile with the approved protocol number 261/2016, and the Universidad de los Lagos with the approved protocol number 001/2023.

### 2.2. Primary Culture Cells

For the primary culture, we obtained small pieces of tissue (approximately 10 mg) or explant of the head and posterior kidney from *O. kisutch* under aseptic or sterile conditions, following Nualart et al. (2023)’s [34] protocol, which was then seeded and kept in a six-well plate and maintained at 18 °C for 24 h under air atmosphere. The cell and tissue growth medium was Leibovitz’s 15 (L-15) supplemented with 10% fetal bovine serum (FBS) (Invitrogen, Thermo Fisher Scientific, Waltham, MA, USA) and 1% penicillin-streptomycin (P/S) (Gibco, Thermo Fisher, Waltham, MA, USA).

### 2.3. In Vitro Treatment

Twenty-four hours after seeding the explant of the head and posterior kidney, primary cell cultures were treated with cortisol 200 ng/mL, tryptophan 5 μg/mL, tryptophan 50 μg/mL, and a combination of cortisol 200 ng/mL + tryptophan 5 μg/mL and cortisol 200 ng/mL + tryptophan 50 μg/mL, using the same concentrations previously described for Castro et al. (2011) [35] and Mardones et al. (2018) [36]. Primary cell cultures in six-well plates were exposed for 1, 3, 6, 12, 24, and 48 h at 18 °C. Control plates had the same volume of medium without the treatment. All experiments were run in triplicate (biological replicates) and were repeated twice independently (as a technical replicate).

### 2.4. Total RNA Extraction

Total RNA from the head and posterior kidney portions, from both stimulated and control tissues, were isolated using TRIzol reagent (Sigma, St. Louis, MO, USA) following the manufacturer’s instructions and stored at −80 °C. Subsequently, RNA was quantified at 260 nm on a NanoDrop spectrophotometer (NanoDrop Technologies^®^, Wilmington, DE, USA), and the quality was determined through electrophoresis on a 1% agarose gel. Finally, total RNA (2 μg) was used as a reverse transcription template to synthesize cDNA, applying MMLV-RT reverse transcriptase (Promega, Madison, WI) and the oligo-dT primer (Invitrogen, Waltham, MA, USA) according to standard procedures.

### 2.5. qRT-PCR Analysis of Gene Expression

Reactions were carried out on an AriaMx Real-time PCR System (Agilent, Santa Clara, CA, USA). CDNA was diluted to 100 ng and used as a qRT-PCR template with reactive Brilliant SYBRGreen qPCR (Stratagene, San Diego, CA, USA). Reactions were performed in triplicate, in a total volume of 14 μL, which contained 6 μL SYBRGreen, 2 μL cDNA, 1.08 μL of primers mix, and 4.92 μL of PCR-grade water. The applied PCR program was as follows: 95 °C for 10 min, followed by 40 cycles at 90 °C for 10 s, 60 °C for 15 s, and 72 °C for 15 s. Melting curve analysis of the amplified products was performed after each PCR to confirm that only one PCR product was amplified and detected. Expression levels were analyzed using the comparative Ct method (2^−ΔΔCT^) [37]. The data are presented as the fold change in gene expression normalized to an endogenous reference gene (*18S*) and relative to unstimulated cells (control). The primers used for *TPH2*, *5-HT 1α receptor*, *INFγ receptor*, *KIAT 2* are listed in Table 1. PCR efficiencies were determined through linear regression analysis of sample data using LinRegPCR [38] from the serial dilutions when Log dilution was plotted against DCT (threshold cycle number).

### 2.6. Statistical Analysis

All statistical analyses and graphs were performed using the software Sigma Plot 11 and Minitab 19. The assumptions of normality, independence, and homogeneity of the residuals for the between-group variances were also tested using a Shapiro–Wilk test and a Levene test, respectively. Significant differences in gene expression between different treatments were determined through three-way analysis of variance (three-way ANOVA), while the factors of variance were the conditions cortisol, tryptophan, and a combination of cortisol + tryptophan in high and low concentrations and times (1–48 h). Tukey tests were used to evaluate the a posteriori difference between groups in relative gene expression (*p* < 0.01), and all data are shown as the mean ± standard error (SE) and represent the relative expression (2^−ΔΔCt^) normalized to a reference gene (*18S*) and compared to the control group at each time point.

## 3. Results

### 3.1. Tryptophan Hydroxylase (TPH2) mRNA Gene Expression Changes

The mRNA gene expression of TPH2 in head kidney cells stimulated with cortisol significantly increased at 3 and 12 h in O. kisutch (Figure 2A). Meanwhile, the gene expression of tryptophan significantly increased at 3 and 48 h in both concentrations. Nevertheless, the combined effect of cortisol and tryptophan (at 5 and 50 μg/mL) presented high levels at 1, 3, and 48 h. However, cortisol and tryptophan at 50 μg/mL also increased their levels at 24 h. Coincidentally, at 6 h, all experimental groups presented a downregulation (Figure 2A and Table 2).

In posterior kidney cells stimulated with cortisol and tryptophan, 5 μg/mL significantly increased at 12 h in O. kisutch (Figure 1B), while tryptophan 50 μg/mL significantly upregulated at 3 and 12 h. The combined effect of cortisol and tryptophan (at 5 and 50 μg/mL) presented high levels at 3, 12, and 48 h (Figure 2B). Likewise, at 1 and 6 h, all experimental groups presented downregulation (Figure 2B and Table 3).

### 3.2. 5HT1α Receptor mRNA Gene Expression Changes

The mRNA gene expression of *5-HT 1α* in head kidney cells stimulated with cortisol significantly increased at 3, 12, and 24 h in *O. kisutch* (Figure 3A and Table 2), while tryptophan stimulation with 5 μg/mL presented an upregulation at 12 h, instead of stimulation with 50 μg/mL which presented an upregulation at 1, 12, and 24 h. The combined effect of cortisol and tryptophan at 5 μg/mL presented an upregulation at 1, 3, and 48 h, while cortisol and tryptophan at 50 μg/mL were highest at 3, 24, and 48 h. 

Posterior kidney cells stimulated with cortisol presented an upregulation at 12 h; instead, tryptophan 5 μg/mL presented an upregulation at 3 and 12 h, while tryptophan 50 μg/mL was highest at 12 h. The combined effect of cortisol and tryptophan 5 μg/mL presented an upregulation at 1, 3, 12, and 48 h, and cortisol and tryptophan 50 μg/mL presented the highest levels at 12 and 24 h (Figure 3B and Table 3).

### 3.3. INFγ Receptor mRNA Gene Expression Changes

The mRNA gene expression of *INFγ* in head kidney cells stimulated with cortisol significantly increased at 3 and 12 h in *O. kisutch* (Figure 4A and Table 2), while tryptophan stimulation with 5 μg/mL did not present changes compared to the control group; instead, stimulation with 50 μg/mL presented upregulation at 3 and 24 h. The combined effect of cortisol and tryptophan at 5 and 50 μg/mL had the highest expression levels at 24 and 48 h.

Posterior kidney cells stimulated with cortisol presented an upregulation at 12 h; instead, tryptophan 5 and 50 μg/mL presented an upregulation at 3 and 12 h. The combined effect of cortisol and tryptophan 5 and 50 μg/mL presented an upregulation at 1, 6, 12, and 48 h (Figure 4B and Table 3).

### 3.4. Kynurenine Aminotransferase 2 (KIAT 2) mRNA Gene Expression Changes

The mRNA gene expression of *KIAT 2* in head kidney cells stimulated with cortisol and tryptophan 5 and 50 μg/mL significantly increased at 12 h in *O. kisutch* (Figure 4A). The combined effect of cortisol and tryptophan 5 μg/mL presented an upregulation at 1, 3, 12, and 48 h, and also cortisol and tryptophan 50 μg/mL presented an upregulation at 12, 24, and 48 h (Figure 5A and Table 2). 

In posterior kidney cells stimulated with cortisol, *KIAT 2* significantly increased at 12 h in *O. kisutch* (Figure 5B), while tryptophan stimulation with 5 μg/mL did not present changes compared to the control group; instead, stimulation with 50 μg/mL presented an upregulation at 3 h. The combined effect of cortisol and tryptophan 5 and 50 μg/mL presented an upregulation at 1, 3, 24, and 48 h; also, cortisol and tryptophan 50 μg/mL presented the highest level at 12 h (Figure 5B and Table 3).

## 4. Discussion

Cortisol is a product of HPI, derived from ACTH activation in the head kidney due to stress events, and tryptophan is an essential amino acid that needs to be incorporated into the diet [1,23,39]. In the last ten years, as recommendations to improve aquaculture have arisen, many articles suggest supplementing the diet with several products, such as tryptophan, since the levels of this essential amino acid are decreasing in fish affected by several stressors, as was mentioned by previous studies [40,41]. Nevertheless, Nile tilapia “*Oreochromis niloticus*” were fed with tryptophan food supplementation but the cortisol levels in tilapia stressed were not reduced [42], and although it reduced glucose levels as a secondary stress marker in Meagre “*Argyrosomus regius*” [41], the lowest growth rate was in fish with tryptophan supplementation.

Nualart et al. (2023) [34] indicated that the primary cell culture in tissue such as the kidney is an excellent approach to obtain results that are more specific than in vivo studies because the cells are not influenced by other tissues. In our study, we used both portions of the kidney, “head kidney and posterior kidney,” which had different responses based on the stimulations. Our results regarding tryptophan hydroxylase (*TPH2*) and *5HT1α* receptor expression of mRNA showed patterns associated with stimulation with tryptophan and tryptophan + cortisol, especially *TPH2* where the high levels were at the earliest and latest points in primary cell cultures in both kidney portions. This enzyme converts tryptophan to serotonin and melatonin as the final products. Both hormones are involved in many physiological processes such as feeding intake, osmoregulation, and reducing stress [1,5,13,43], indicating that the serotonin system could have a dual role in the stress response, responding as an early signal during initiation but also as a late response during chronic stress as we mentioned in our results with two patterns of mRNA expression.

The IFN-γ cytokine is produced by natural killer cells (NK cells) and T lymphocytes to avoid virus replications [44]. The *IFNγ* in our study presented the highest levels of mRNA expression in the posterior kidney than the head kidney, and its response was highest in all of the kinetic time points except at 24 h. Tryptophan stimulation increased IFNγ expression levels in both tissues, but both stimulators were stimulated at the latest time points, which is contrary to the results obtained by the previous studies [45,46], where glucocorticoid and cortisol suppressed the immune response and, of course, IFNγ production, weakening the response against viruses. Saravia et al. (2022) [47] in *Harpagifer antarcticus* indicated that both immunostimulants (LPS and Poly:C) were overexpressed, suggesting that interferon could play a role in antigen presentation in both types of infection for Antarctic Notothenioids.

Some authors [48] indicated that tryptophan is metabolized via the kynurenine pathway (see Figure 1). However, this pathway has two sub-pathways after kynurenine appears in two potential ways, quinolinic acid and kynurenic acid, where the latter has a kynurenine aminotransferase 2 (*KIAT 2*) enzyme that metabolizes kynurenic acid. Our results of *KIAT 2* mRNA expression showed that cortisol and tryptophan stimulate its overexpression, especially at the latest time point of the experiment [32]. Moreover, Badawy (2017) [30] indicated that IFNγ and glucocorticoids induce the expression of indoleamine 2,3-dioxygenase “IDO,” which catalyzes the same step as tryptophan 2,3-dioxygenase “TDO” in the kynurenine pathway. Therefore, we stimulated with cortisol and tryptophan; as mentioned previously, IFNγ was overexpressed in the latest experimental time points; consequently, the kynurenine pathway can be induced, having high levels of KIAT2. Also, the IFNγ-stimulation of macrophages has been closely linked to inducing the mammalian macrophage IDO response [49]. *Lampetra japonicum* (lamprey) demonstrated that exogenous tryptophan could inhibit the expression of pro-inflammatory factors and promote the expression of anti-inflammatory factors by inhibiting the MyD88 signaling pathway and activating the IDO-KYN-AHR (aryl hydrocarbon receptor) signaling pathway [48]. In male BALB/c mice [50], exogenous kynurenic acid (KYNA) was indicated to have low levels of cytotoxicity toward murine splenocytes and exhibited immunotropic properties. In rainbow trout (*Oncorhynchus mykiss*), it was demonstrated that cortisol increased kynurenine levels at 48 h post-stress events in the liver and brain [51], which are in line with our results. In addition, in rainbow trout, the kidney was shown to present necrosis in epithelia, tubules, and glomerulus in practically all KYNA-supplemented groups; similar problems were presented in gills with fusion or disappearance of gill lamellae, of course affecting the osmoregulation and respiratory activity [33,52]. The TRP–KYN pathway in lamprey leukocytes (*Lampetra japonica*) was activated by adding TRP [53] and overexpression significantly reduced TNF-α and NF-κβ expression and clarified the ancestral features and functions of the TRP–KYN pathway. Meanwhile, Machado et al. (2021) [54] indicated that the tryptophan presented a clear role in the tolerance process responsible for restriction of the pro-inflammatory cluster of the immune response in Head Kidney Leucocytes (HKLs) of European seabass. 

## 5. Conclusions

Supplementing tryptophan and cortisol in primary cell culture can be an excellent way to discover how the different pathways act. Le Floc’h et al. (2011) [55] and Michael et al. (1964) [56] indicated that tryptophan is degraded through three metabolic pathways, with 95% being degraded through the kynurenine pathway, indicating the relevance of knowing how this pathway is activated or not, and if stress levels associated with fish culture trigger its activation. It would also be essential to know in different species what the consequence is of increasing KYNA levels in the short and long term and even during the ontogeny of the fish.

## Figures and Tables

**Figure 1 animals-13-03562-f001:**
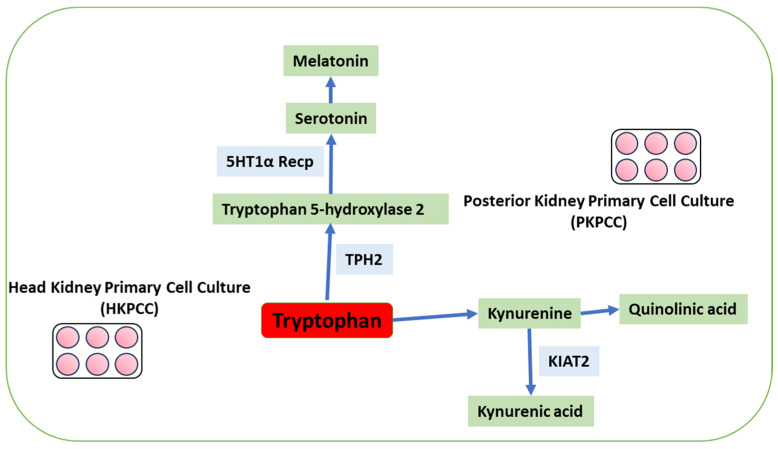
Diagram of the kynurenine pathway from tryptophan. Blue arrows show the product in each box. Light blue boxes are enzymes or receptors of this pathway, previous to the final product.

**Figure 2 animals-13-03562-f002:**
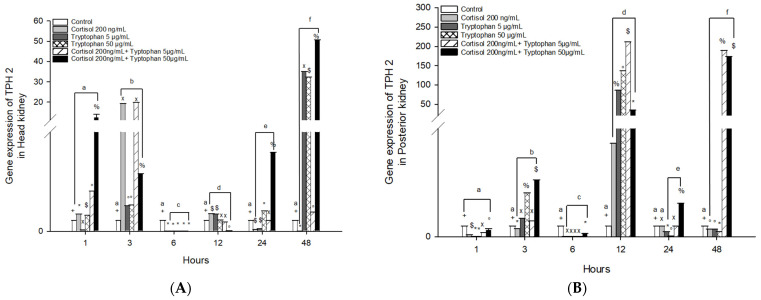
Gene expression of *TPH2* in the (**A**) head kidney primary cell culture (HKPCC) and (**B**) posterior kidney primary cell culture (PKPCC) of *O. kisutch* treated with cortisol, tryptophan, and a combination of cortisol and tryptophan at high and low concentrations. Each value represents the mean ± S.E.M. (*n* = 3). Different letters indicate statistical differences within the same treatment between time points. Symbols (*, °, %, x, $, and +) indicate statistical differences between different treatments (control, cortisol, and tryptophan) at the same time point (three-way ANOVA, *p* < 0.01).

**Figure 3 animals-13-03562-f003:**
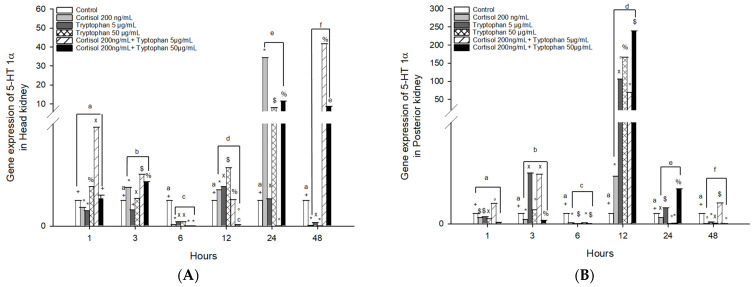
Gene expression of *5HT 1α* in the (**A**) head kidney primary cell culture (HKPCC) and (**B**) posterior kidney primary cell culture (PKPCC) of *O. kisutch* treated with cortisol, tryptophan, and a combination of cortisol and tryptophan in high and low concentrations. Each value represents the mean ± S.E.M. (*n* = 3). Different letters indicate statistical differences within the same treatment between time points. Symbols (*, °, %, x, $, and +) indicate statistical differences among different treatments (control, cortisol, and tryptophan) at the same time point (three-way ANOVA, *p* < 0.01).

**Figure 4 animals-13-03562-f004:**
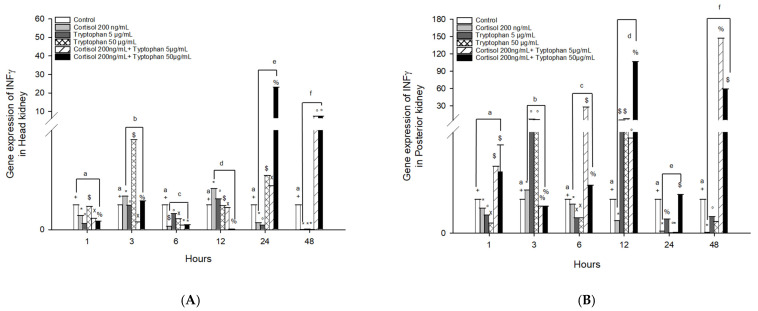
Gene expression of *IFNγ* in the (**A**) head kidney primary cell culture (HKPCC) and (**B**) posterior kidney primary cell culture (PKPCC) of *O. kisutch* treated with cortisol, tryptophan, and a combination of cortisol and tryptophan at high and low concentrations. Each value represents the mean ± S.E.M. (*n* = 3). Different letters indicate statistical differences within the same treatment between time points. Symbols (*, °, %, x, $, and +) indicate statistical differences between different treatments (control, cortisol, and tryptophan) at the same time point (three-way ANOVA, *p* < 0.01).

**Figure 5 animals-13-03562-f005:**
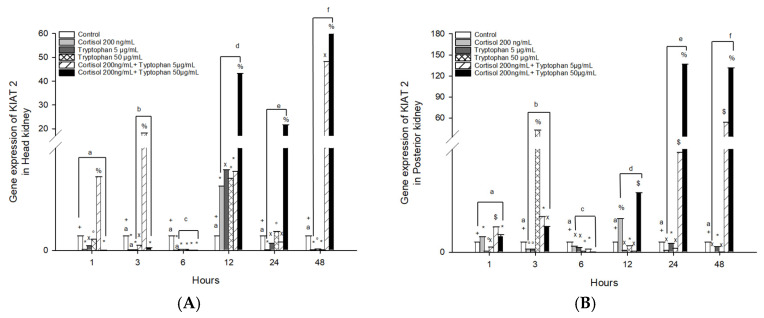
Gene expression of *KIAT 2* in the (**A**) head kidney primary cell culture (HKPCC) and (**B**) posterior kidney primary cell culture (PKPCC) of *O. kisutch* treated with cortisol, tryptophan, and a combination of cortisol and tryptophan in high and low concentrations. Each value represents the mean ± S.E.M. (*n* = 3). Different letters indicate statistical differences within the same treatment between time points. Symbols (*, °, %, x, $, and +) indicate statistical differences among different treatments (control, cortisol, and tryptophan) at the same time point (three-way ANOVA, *p* < 0.01).

**Table 1 animals-13-03562-t001:** Primer sequences.

Primer	Nucleotide Sequences (5′→3′)	Efficiency Head Kidney (%)	Efficiency Posterior Kidney (%)	
*18S Fw*	GTCCGGGAAACCGTC	101.9	100.5	XR_006760234.1
*18S Rv*	TTGAGTCAAATTAAGCCGCA			
*TPH 2 Fw*	AGTGTAGCTCAGTGGAGGA	101.4	103.2	XM_014125607.2
*TPH 2 Rv*	AATGCACTGGAGAGGATGTT			
*INF γ receptor Fw*	ATCGCTCCCTATTTCTCTGTG	99.1	100.2	NM_001360942.1
*INF γ receptor Rv*	CCAAGACACCCAACAGGAT			
*KIAT 2 Fw*	TGCACAGCGGAGAAGGTACAGTGG	101.6	104.8	XM_045693024.1
*KIAT 2 Rv*	GGCTCCGACAGTGACCAGGATGT			
*5-HT 1α receptor Fw*	TGGAGTGCTCAGTGACTGGT	97.4	100.2	XM_014173861.2
*5-HT 1α receptor Rv*	AGCCCTTTAGTCCAGCCTCTAC			

**Table 2 animals-13-03562-t002:** *p*-values from three-way ANOVA, with the interaction of parameters (condition*time) in the head kidney primary cell culture (HKPCC) of *O. kisutch* treated with cortisol, tryptophan, and a combination of cortisol and tryptophan in high and low concentrations. Statistical differences between different treatments (control, cortisol, and tryptophan) at the same time point, condition, and times are the main factors. NS, not significant; significant (*p* < 0.01).

Tissue	Genes	Times	Control	Cortisol 200 ng/mL	Tryptophan 5 μg/mL	Tryptophan 50 μg/mL	Cortisol 200 ng/mL + Tryptophan 5 μg/mL	Cortisol 200 ng/mL + Tryptophan 50 μg/mL
**Head Kidney**	**5HTP1a**	**1**	NS	<0.01	<0.01	<0.01	<0.01	<0.01
		**3**	NS	<0.01	<0.01	<0.01	<0.01	<0.01
		**6**	<0.01	NS	<0.01	<0.01	NS	NS
		**12**	NS	<0.01	<0.01	<0.01	<0.01	<0.01
		**24**	<0.01	<0.01	<0.01	<0.01	NS	<0.01
		**48**	<0.01	NS	<0.01	NS	<0.01	<0.01
**Head Kidney**	**INFg**	**1**	NS	<0.01	NS	<0.01	<0.01	NS
		**3**	<0.01	<0.01	<0.01	<0.01	NS	<0.01
		**6**	NS	NS	<0.01	NS	NS	NS
		**12**	<0.01	<0.01	<0.01	<0.01	<0.01	NS
		**24**	<0.01	NS	NS	<0.01	<0.01	<0.01
		**48**	<0.01	NS	NS	NS	<0.01	<0.01
**Head Kidney**	**KIAT**	**1**	NS	NS	<0.01	<0.01	<0.01	NS
		**3**	NS	NS	NS			NS
		**6**	NS	NS	NS	NS	NS	NS
		**12**	<0.01	<0.01	<0.01	<0.01	<0.01	<0.01
		**24**	<0.01	NS	<0.01	<0.01	<0.01	<0.01
		**48**	<0.01	NS	NS	NS	<0.01	<0.01
**Head Kidney**	**TPH2**	**1**	<0.01	<0.01	<0.01	<0.01	<0.01	<0.01
		**3**	<0.01	<0.01	<0.01	<0.01	<0.01	<0.01
		**6**	<0.01	<0.01	<0.01	<0.01	<0.01	<0.01
		**12**	<0.01	<0.01	<0.01	<0.01	<0.01	<0.01
		**24**	<0.01	<0.01	<0.01	<0.01	<0.01	<0.01
		**48**	<0.01	<0.01	<0.01	<0.01	<0.01	<0.01

**Table 3 animals-13-03562-t003:** *p*-values from three-way ANOVA, with the interaction of parameters (condition*time) in the posterior kidney primary cell culture (PKPCC) of *O. kisutch* treated with cortisol, tryptophan, and a combination of cortisol and tryptophan in high and low concentrations. Statistical differences between different treatments (control, cortisol, and tryptophan) at the same time point, Condition, and Times are the main factors. NS, not significant; significant (*p* < 0.01).

Tissue	Genes	Times	Control	Cortisol 200 ng/mL	Tryptophan 5 μg/mL	Tryptophan 50 μg/mL	Cortisol 200 ng/mL + Tryptophan 5 μg/mL	Cortisol 200 ng/mL + Tryptophan 50 μg/mL
**Posterior Kidney**	**5HTP1a**	**1**	NS	<0.01	<0.01	<0.01	<0.01	NS
		**3**	NS	NS	<0.01	<0.01	<0.01	NS
		**6**	NS	<0.01	NS	NS	NS	NS
		**12**	<0.01	<0.01	<0.01	<0.01	<0.01	<0.01
		**24**	NS	<0.01	<0.01	NS	<0.01	NS
		**48**	NS	NS	NS	NS	<0.01	<0.01
**Posterior Kidney**	**INFg**	**1**	<0.01	<0.01	<0.01	<0.01	<0.01	<0.01
		**3**	<0.01	<0.01	<0.01	<0.01	<0.01	<0.01
		**6**	<0.01	<0.01	<0.01	<0.01	<0.01	<0.01
		**12**	<0.01	<0.01	<0.01	<0.01	<0.01	<0.01
		**24**	<0.01	<0.01	<0.01	<0.01	<0.01	<0.01
		**48**	<0.01	<0.01	<0.01	<0.01	<0.01	<0.01
**Posterior Kidney**	**KIAT**	**1**	NS	<0.01	NS	<0.01	<0.01	<0.01
		**3**	NS	<0.01	<0.01	<0.01	<0.01	<0.01
		**6**	NS	<0.01	<0.01	NS	<0.01	NS
		**12**	<0.01	<0.01	NS	<0.01	NS	<0.01
		**24**	<0.01	NS	<0.01	<0.01	<0.01	<0.01
		**48**	<0.01	NS	<0.01	<0.01	<0.01	<0.01
**Posterior Kidney**	**TPH2**	**1**	NS	NS	NS	NS	<0.01	<0.01
		**3**	NS	<0.01	<0.01	<0.01	<0.01	<0.01
		**6**	NS	NS	NS	NS	NS	NS
		**12**	NS	<0.01	<0.01	<0.01	<0.01	<0.01
		**24**	NS	<0.01	<0.01	NS	<0.01	<0.01
		**48**	NS	<0.01	<0.01	<0.01	<0.01	<0.01

## Data Availability

Data availability statements are available by requirement.

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
