# Peer review of "Tryptophan and Cortisol Modulate the Kynurenine and Serotonin Transcriptional Pathway in the Kidney of Oncorhynchus kisutch"

_animals, 2023, doi:10.3390/ani13223562_

Round 1

Reviewer 1 Report

Comments and Suggestions for Authors

Dear editor,

This manuscript of Tryptophan and Cortisol are modulating the Kynurenine and Serotine transcriptional pathway in kidney of Oncorhynchus kisutch ” submitted by Luis Vargas-Chacof,et al., indicated how the supplement of Cortisol, Tryptophan and a combination of Cortisol and Tryptophan in high and low concentrations in primary cells in the head and posterior kidneys activated Kynurenine and Serotine transcriptional pathway. However, there is no new insight in this manuscript and no clear way to know the relationship between experiment and production. For this reason, I suggest to improve the manuscript before publication. The comments are as following:

1) The English writing should be carefully checked and polished. For example, in 2.6. Statistical Analysis part, write “reference gene” incorrectly as “reference gen”.  

2) The references mentioned in the background are almost all results obtained through aquaculture experiments, but your experiments focuses on cell experiments, so I don't think the persuasiveness of these references is particularly strong. Could you please supple the research progress of related studies at the cellular level? At the same time, literature citations are unclear, please avoid too many references at once like [19-26].

3)   How was the gene sequence obtained? Please provide the literature source or gene number of the gene sequence.

4) Can you provide a more detailed description of the process of cell culture? Because the state of each fish is different, if the primary cells of different treatment groups come from different fish, I think the results obtained from this experiment are not meaningful.

5) I found that there were only three replicates in each treatment group in your PCR experiment, the statistical significance of three replicates is not strong enough, so please increase the number of replicates.

6)  If the two metabolic pathways introduced in the core of this paper can be shown in the form of a diagram, the context of the article will be more clear.

Comments on the Quality of English Language

7)       The discission part should be re-written to improve the readability and logicality. The authors should discuss the inner mechanism really based on the data obtained in this study, so as the conclusions part. Please give a brief description and highlights of the test results when conclude.

Reviewer 2 Report

Comments and Suggestions for Authors

The presentation of data and the way of writing need so much effort and are far away from a standard scientific manuscript. Further, some data does not look alright, and they need to check all data again. The statistical analysis should be clearer with reporting the interaction effect as well. If the interaction was significant, you can unpack original data. If the interaction was not significant, you can analyze the main effect. However, if you consider the effect of time as well can be three-way ANOVA!! Due to these issues, I did not go through line-by-line comments.

Comments on the Quality of English Language

Moderate improvment

Reviewer 3 Report

Comments and Suggestions for Authors

The full scientific name of the fish species should be included in the title.

The simple summary can be reconsidered.

"The aquaculture fish are kept long in sea cages or tanks. " incomplete sentence and make non-sense here.

The abstract section should briefly introduce the research background and research significance and clarify the research methods, then introduce the main research results, and finally, give the corresponding conclusions. The abstract of this article looks ok, but some descriptions are redundant, and I hope to make a comprehensive modification. Also, there is no mention of the trial methods or conditions. 

Line 34: Why " It is known that " in the abstract. Revise and tell the study results rather than telling results from the literature.

Line 38: Before using abbreviations, the full definitions should be mentioned first. 

Line 44: "The last time the fish in aquaculture" What does this mean? Actually the whole manuscript must be revised by a native English speaker to avoid such mistakes.

Line 50: The term of "aquaculture fish" is repeated several times in the manuscript. It is not common in the field of fish farming, you may say "aquatic animals" or farmed fish".

Line 50-51: need relevant refs. 

Line 58-60: why these lines separated from the previous paragraph? same sequence should be. 

Line 63-64: this repeats what has been mentioned earlier in lines 44-48.

Line 68-69: consider italic formatting and check elsewhere too.

Line 81, 106, 115, 239, 240, 246 and elsewhere: consider the correct formatting for citations according to the journal guidelines.

Line 85: befor te objectives, I suggest to elaborate on Coho salmon (Oncorhynchus kisutch) and its origin, commercial value, ... etc. Also what are the common stressors facing this fish species and the expected response. 

Line 94: can you include the fish length too?

Line 96: did you start sampling before adapting fish to the lab.? I think this should be another stressful factor? 

Line 100-101: ethical code NO.?

Line 105: O. kisutch, italic

Line 123, 136, 150 and elsewhere: Make sure that symbols, sub-, and super-scripts, upper- and lower-case are presented correctly and that there is the correct and consistent use of italics, brackets, and punctuation, etc.

Line 138 and elsewhere: I think genes can be italics.

Table 1: Did the authors perform a stability analysis to select the most stable reference genes? also the accession No. is needed

Line 298-302: remove from the abstract. The section needs to revise. The object of the conclusion is not decided.

There are mistakes in the reference list, including incorrect italics with the use of punctuation, etc.

Comments on the Quality of English Language

 Extensive editing of the English language required

Round 2

Reviewer 1 Report

Comments and Suggestions for Authors

The authors have addressed all my concerns, and the manuscript is suitable for publication in this journal.

Comments on the Quality of English Language

The authors have addressed all my concerns, and the manuscript is suitable for publication in this journal.

Author Response

Dear Editor 

Thank you very much.

Sincerely,

Dr. Luis Vargas-Chacoff

Reviewer 2 Report

Comments and Suggestions for Authors

Unfortunately, the authors could not improve the MS and it still is far away from the standard level.

Comments on the Quality of English Language

Fixing some errors

Author Response

(The authors gave the same response as above.)

Reviewer 3 Report

Comments and Suggestions for Authors

The manuscript is well-revised but still some minor revisions are required. Make sure that symbols, sub-, and super-scripts, upper- and lower-case are presented correctly and that there is the correct and consistent use of italics, brackets, and punctuation, etc. See lines 47-48, 81, 87, 145, 155, 164, 179, 195, 196, 235, 265, 297, 328, 334, 350, 368, 369, 372, 373, 374, 392-394, 398, 404, 410, 414, 417-419, ... and elsewhere.

All genes should be formatted in italics. text and table 1.

No references should be written in the conclusion. 

Author Response

Dear Reviewer 3
